# Endemic infectious cutaneous ulcers syndrome in the Oti Region of Ghana: Study of cutaneous leishmaniasis, yaws and *Haemophilus ducreyi* cutaneous ulcers

**Richard Adjei Akuffo**[1]*, **Carmen Sanchez**[2], **Ivy Amanor**[1], **Jennifer Seyram Amedior**[1], **Nana Konama Kotey**[3], **Francis Anto**[4], **Thomas Azurago**[3], **Anthony Ablordey**[1], **Felicia Owusu-Antwi**[5], **Abate Beshah**[6], **Yaw Ampem Amoako**[7], **Richard Odame Phillips**[7], **Michael Wilson**[1], **Kingsley Asiedu**[8], **Jose-Antonio Ruiz-Postigo**[8], **Javier Moreno**[2], **Mourad Mokni**[9]

**1** Noguchi Memorial Institute for Medical Research, University of Ghana, Accra, Ghana, **2** WHO Collaborating Center for Leishmaniasis, Instituto de Salud Carlos III, CIBERINFEC, Madrid, Spain, **3** Ghana Health Service, Accra, Ghana, **4** School of Public Health, University of Ghana, Accra, Ghana, **5** Ghana Country Office, World Health Organization, Accra, Ghana, **6** World Health Organization Regional Office for Africa, Brazzaville, Republic of Congo, **7** Kumasi Centre for Collaborative Research, Kwame Nkrumah University of Science and Technology, Kumasi, Ghana, **8** Department of Control of Neglected Tropical Diseases, World Health Organization, Geneva, Switzerland, **9** La Rabta Hospital Dermatology Department, Research Laboratory, Faculty of Medicine, University of al-Manar 2, Tunis, Tunisia

* richard.akuffo@gmail.com

**Data Availability Statement:** All relevant data are within the paper and its Supporting information files.

## Abstract

### Background

A recent study detected cutaneous leishmaniasis (CL) in 31.9% of persons with skin ulcers in the Oti Region of Ghana, resulting in a need to investigate other potential causes of the unexplained skin ulcers.

### Methodology/Principal findings

A community based cross-sectional study was conducted in the Oti region to investigate skin ulcers of undetermined aetiologies. To confirm a diagnosis of cutaneous leishmaniasis, Buruli ulcer, *Haemophilus ducreyi* ulcers, or yaws, DNA obtained from each patient skin ulcer sample was systematically subjected to polymerase chain reaction (PCR) for *Leishmania spp.*, *Mycobacterium ulcerans*, *Haemophilus ducreyi*, and *Treponema pallidum* sub species *pertenue*. A total of 101 skin ulcer samples were obtained from 101 persons. Co-infection of more than one organism was observed in 68.3% of the samples. Forty (39.6%) participants had a positive result for *Leishmania spp.*, 68 (67.3%) for *Treponema pallidum* sub. Sp. *pertenue*, and 74 (73.3%) for *H. ducreyi*. Twenty (19.8%) of the patient ulcers were simultaneously infected with *Leishmania spp.*, *Treponema pallidum* sub. Sp. *pertenue*, and *H. ducreyi*. None of the patients' lesions yielded a positive result for *Mycobacterium ulcerans*.

**Funding:** The author(s) received no specific funding for this work.

**Competing interests:** The authors have declared that no competing interests exist.

**Abbreviations:** BU, Buruli ulcer; CL, Cutaneous leishmaniasis; HD, *Haemophilus ducreyi*; WHO, World Health Organization.

## Conclusions/Significance

This study detected single and mixed occurrence of the causative organisms of CL, yaws, and *H. ducreyi* cutaneous ulcers in CL endemic communities of the Oti Region in Ghana. These findings emphasize the importance of integrating multiple skin diseases on a common research platform and calls for the development of a comprehensive guideline for diagnosing and treating tropical ulcers in the study areas.

## Introduction

It is estimated that skin conditions affect more than 900 million people worldwide, ranking as the 15th leading cause of years of healthy life lost due to disability annually [1–3]. The World Health Organization launched the strategic framework for integrated control and management of skin-related neglected tropical diseases (skin NTDs) to give new impetus to the clarion call for integrated skin diseases research [4]. The skin NTDs include Buruli ulcer, cutaneous leishmaniasis (CL), leprosy, mycetoma, yaws, onchocerciasis and lymphoedema [5]. Other skin conditions which are not part of the skin NTDs but identified as cause of ulcers include *Haemophilus ducreyi* [6].

Generally, infectious cutaneous ulcers (ICU) are a major medical problem in low-resource tropical countries of Africa and the South Pacific where they are estimated to affect approximately 100,000 children each year [7]. While the cause of two-thirds of ICUs in the tropics have largely been attributable to *Treponema pallidum* Subspecies *Pertenue* and *Haemophilus ducreyi*, the cause(s) of the remaining one-third is often unknown or unclassified [6–10]. The detection of *H. ducreyi* as a cause of non-genital skin ulcers in some countries including Ghana, especially in yaws endemic areas, is of particular interest given that *H. ducreyi* had for many years only been associated with the sexually transmitted disease chancroid [6, 11–14].

As several skin ulcers present with similar lesion characteristics, determination of the underlying aetiology on clinical grounds only can be a challenge, especially in areas having multiple causes of similar skin ulcers. For instance, the primary lesions of yaws have been described as being similar to CL, or tropical ulcers including those caused by Fusobacteria. Recent studies have indicated difficulty in clinically distinguishing ICU due to yaws from those attributable to *H. ducreyi* [12, 15, 16]. Another skin ulcer considered endemic in parts of Ghana is Buruli ulcer [17–19].

There is an urgent need for studies to unravel the aetiologies and the proportion of yaws-like skin ulcers that remain unexplained especially in endemic areas if the goal of eradicating yaws by 2030 is to be achieved [20]. For instance, in situations where skin ulcers from some yaws endemic areas in Ghana were systematically screened for confirmation of yaws, *H. ducreyi*, and Buruli ulcer, a significant number of the ulcers remained negative and unexplained [9, 21]. Some recent studies have detected other bacterial infections as potential causes of some of the unexplained ulcers, and in some instances, classified ulcers as polymicrobial ulcers [7, 12].

There has however been paucity of data on CL research especially in yaws endemic areas of Africa and the South Pacific, which have indicated large proportions of unexplained ulcers [9, 22]. A localized outbreak of skin ulcers suspected to be cases of CL was first reported in Ghana from the Ho municipality of the Volta Region in 1999 based on the identification of *Leishmania* amastigotes in some human skin lesion biopsies examined by microscopy [23].

Subsequent studies in the Volta Region suggest a rather complex nature of the CL in the region. The DNA of *Leishmania major* was detected in CL patients sampled from the Ho

municipality of the Volta Region in 2002 [24]. A follow-up study involving biopsies from CL cases sampled in 2006 and 2007 from Taviefe, a community located about 10km north of the Ho municipality of the Volta Region, did not confirm *L. major*. Instead, an uncharacterized species of *Leishmania* was detected [25].

An additional study in some other parts of the Ho municipality successfully cultured and obtained three isolates from individuals suspected with active CL. Using DNA sequencing and phylogenetic analysis, the isolates were confirmed to be part of the *Leishmania enriettii* complex, a new subgenus of *Leishmania* parasites [26]. *L. tropica*, and *L. major* DNA were identified in *Sergentomyia* sand flies sampled from the Ho municipality [27]. Identification of *L. major*, uncharacterized *Leishmania* species, and recently, members of the *Leishmania enriettii* complex from CL cases in the Ho municipality suggests a complex epidemiology of CL in the region.

The Oti Region was part of the Volta Region of Ghana until 2019 when the then Volta Region was divided into the now Volta and Oti Regions respectively. The portion of the Volta Region now classified as Oti Region had no previous documentation of *Leishmania* infection, prior to 2018. However, parts of the Oti Region have been known to be endemic for yaws [21, 28]. In 2018, we confirmed CL in 32% of persons with skin ulcers in a study conducted in the Oti Region of Ghana [29]. The etiology of skin ulcers in the remaining 68% of persons was not identified in that study.

We therefore hypothesized that yaws, *H. ducreyi* and Buruli ulcer may account for a significant proportion of the over 65% of skin ulcers which tested negative for *Leishmania* infection in our previous study. Hence, we proposed to systematically investigate ulcers from five CL endemic communities in the Nkwanta South and North Districts of the Oti Region of Ghana for *Leishmania* parasite (for CL), *Treponema pallidum* sub species *pertenue* (for yaws), *Mycobacterium ulcerans* (for Buruli ulcer), and *Haemophilus ducreyi* to determine the proportions of ulcers attributable to the respective organisms.

## Materials and methods

### Ethics statement

Ethical approval to conduct this study was obtained from the ethics review committee of the Ghana Health Service (GHS-ERC006/08/18). Written informed consent was obtained from all study participants. For participants under 18 years, written consent was obtained from a parent or guardian, and assent from children 10–17 years old.

### Confidentiality protection

Once consented, all study participants were assigned a unique identifying number. The consent form and the first page of the case report form (CRF) containing both the patient's name and unique identifying number were stored by the Principal Investigator (PI), in locked file cabinets. Apart from the screening and enrolment log, the remaining CRF pages contained only the subject identification number, but not their name. Study personnel involved in data entry and analysis did not have access to participant names or other identifying information, assuring the privacy and confidentiality of the participants. The CRF was the only source document for this study.

### Study design

Using a cross-sectional study design, this study was conducted in five communities of the Oti region of Ghana from 11th to 21st September, 2019.

## Study area

This study was conducted in the following five communities: Ashiabre, Keri, and Dawa in Nkwanta South Municipality and Sibi Hilltop and Obunja in Nkwanta North District.

The population of Nkwanta South municipality is estimated to be 117,878 with males constituting 49.6%. Covering a land area of approximately 2733 km$^2$, the Nkwanta South municipality is located between latitudes 7° 30' and 8° 45' North and longitude 0° 10' and 0° 45'East [30].

The population of the Nkwanta North district is estimated to be 64,553 with males constituting 50.2%. The district is located between latitude 7°30'N and 8°45'N and longitude 0°10'W and 045'E. It shares boundaries with Nkwanta South municipality to the south, Nanumba South to the north, Republic of Togo to the east, and Kpandai District to the west [31].

## Inclusion criteria

Eligible study participants were residents in the study community for $\geq$ 12 months, aged 2 to 75 years and having atraumatic ulcers (symptomatic for >1 week).

## Sample size consideration

All persons with moist atraumatic ulcers in the five CL endemic communities were invited to a central location to participate in the study. The invitation was done through a community-wide announcements information system.

## Sampling method

All persons with moist atraumatic skin ulcers presenting to the screening centre were invited to participate in the study after informed consent was obtained.

## Field procedures

Upon providing consent to participate in the study, an additional consent was obtained to capture images of the participants and their ulcers for presentation and publication purposes. As a result, the option was always available for the participants to freely participate in the survey and also, to freely decline to participate in the survey entirely, or in aspects of the survey. All persons with skin ulcers were screened with a pictorial guide to document the location of the ulcer(s) after which they were all screened in the field for yaws.

From one ulcer per participant, three swabs and a 3mm punch biopsy were obtained. The swabs were obtained by ensuring that the cotton tip of the swab was pressed and rolled over from end to end of the ulcer twice, for each case. The swabs were placed in lysis buffer and stored at 4 degrees Celsius for transportation to the laboratory for processing. The biopsy samples were stored in sterile saline solution and also transported at 4 degrees to the laboratory for processing.

If more than one ulcer (similar in size) was observed on a participant, then only one was randomly selected for sampling. In the event of different ulcer sizes on a given participant, the largest ulcer was sampled. Given that multiple samples (swabs and a punch biopsy) were obtained from the sampled ulcer, the decision to sample only one ulcer was also to make the sampling procedure more tolerable to the participants. A possible downside of this sampling procedure however, is the likelihood of missing discordant ulcers due to different skin ulcer aetiologies on the same participants. Very dry, crusty, or large chronic ulcers > 5 years in duration were however not included in the sampling.

## Laboratory procedures

The biopsy samples were independently tested for *Leishmania* parasite at the WHO collaborating center for leishmaniasis at the Instituto de Salud Carlos III in Spain using PCR techniques according to previously published procedure [32]. Briefly, DNA extraction was performed using SpeedTools Tissue DNA Extraction Kit (Biotools, Inc). A nested polymerase chain reaction (Ln-PCR) approach was used to amplify DNA of *Leishmania* species from the human skin lesions following an adaptation of the protocol by Cruz et al., 2002 [33], with the target being the small subunit ribosomal ribonucleic acid (SSU rRNA) gene. Positive control used was *Leishmania infantum* (JPC strain) with distilled water as negative control (Fig 1).

A set of lesional swabs were independently tested using real time PCR methods as described by Fyfe et al. (2007) [34] for detection of *Mycobacterium ulcerans*, the causative agent for Buruli ulcer, at the Buruli ulcer laboratory of the Noguchi Memorial Institute for Medical Research, University of Ghana (UG-NMIMR). In summary, IS*2404*/internal positive control (IPC) mixtures contained 1 mL of template DNA, 0.9 mM of each primer, 0.25 mM of the probe, 1 x TaqMan$^R$ Universal PCR Master Mix (Applied Biosystems, Foster City, CA), and TaqMan exogenous IPC reagents (Applied Biosystems) in a total volume of 25 mL. Detection was performed on a 7300 real-time PCR System (Applied Biosystems) using the following thermal profile: one cycle of 50 1C for 2 min, one cycle of 95 1C for 10 min, and 40 cycles of 95 1C for 15 s and 60 1C for 1 min. Triplicate positive/negative PCR controls, positive/negative extraction controls, and fluorescence controls were included in each assay. Cycle threshold (CT) values >35 cycles were considered negative [35, 36]. A second set of lesional swabs were also independently tested for both *Treponemal Pallidum* sub species *pertenue* (the causative organism for yaws) and *Haemophilus ducreyi* using a multiplex PCR assay for simultaneously detecting *Treponema pallidum* sub species *pertenue* and *Haemophilus ducreyi* at the yaws testing laboratory at the UG-NMIMR [11]. According to manufacturer guidelines, an Applied Biosystems™ 7500 Real-Time PCR System was used to conduct the multiplex qPCR test using the RealCycler® universal kit (Progenie molecular, Valencia, Spain) for the identification of the *PolA* and *HhdA* specific genes for T. pallidum and H. ducreyi, respectively. Briefly, 14 μL of RealCycler® Universal AmpliMix (a qPCR mastermix reagent) was pipetted into reaction tubes with 6 μL of the DNA template producing 20 μL by mixing. For every run, there were

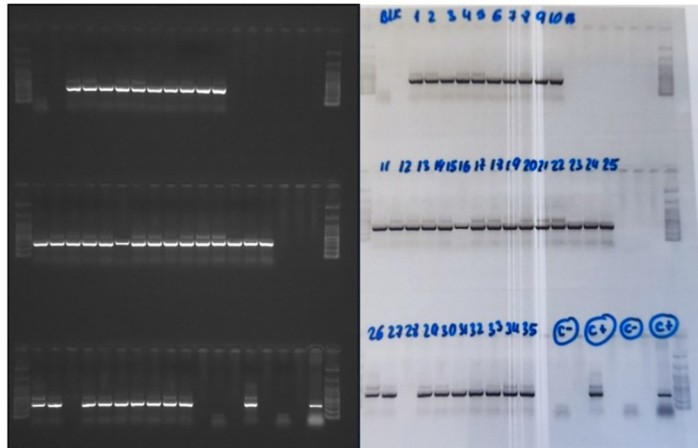

**Fig 1. Visualization of Ln-PCR amplification products on 1.5% agarose gel (C- is negative control; C+ is positive control).** Reprinted under a CC BY license, with permission from Dr Richard Akuffo.

positive and negative controls. The PCR's cycling conditions were observed as required. One (1) cycle of initial denaturing at 95˚C for 15 min, 45 cycles of 95˚C (denaturing) for 5 s, 60˚C (annealing) for 30 s and 72˚C (extension) for 30 s were performed.

### Data management and analysis

Data were managed using Microsoft Access software version 2013 and analyzed using STATA software version 14. All statistical tests were performed at a 95% confidence level. Categorical data were analyzed using the Chi-square test of association and the Fisher's exact test where cell counts were below 5. The Kruskal Wallis H test was used to assess differences in average ulcer sizes across the ulcer categories.

### Results

Of 119 participants with skin ulcers screened, 101 persons with moist, exudative ulcers were included in the study (Fig 2). Eighteen persons were not included in the study because their ulcers were dry, crusty, or re-epithelializing.

The majority (61.4%) of participants were males and mean age (range) for included participants was 13.4 (2–74 years) (Table 1). Table 1 shows that 97% of lesions were located on the lower limbs. Seventy-six (75.2%) of the ulcers had period of onset not exceeding twelve weeks at the time of sampling. The number of ulcers per participant ranged from 1 to 5 with the average ulcer size being 1.7cm (range:0.35–9.2cm). Sixty-nine (68.3%) had a single ulcer while 20 (19.8%), 7 (6.9%), 4 (4.0), and 1 (1.0%) had two, three, four, and five ulcers respectively.

Table 2 shows that 40 (39.6%) out of 101 biopsy samples were positive for *Leishmania* spp. (CL). Sixty-eight (67.3%) samples were positive for *Treponema pallidum* sub. Sp. *Pertenue*

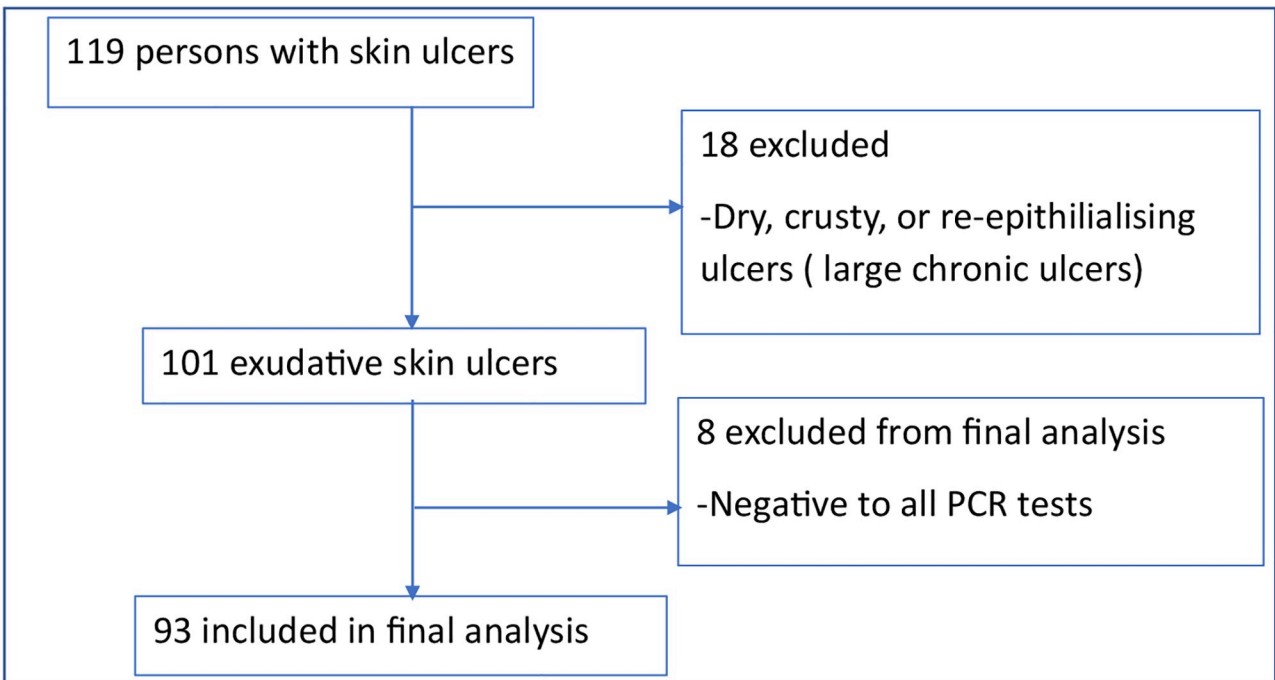

**Fig 2. Algorithm for skin ulcer sampling and analysis.** Reprinted under a CC BY license, with permission from Dr. Richard Akuffo.

**Table 1. Characteristics of participants with skin ulcers.**

| | Communities | | | | | |
|---|---|---|---|---|---|---|
| | **Dawa Akura** | **Ashiabre** | **Keri** | **Sibi Hilltop** | **Obunja** | **Total** |
| | **N (%)** | **N (%)** | **N (%)** | **N (%)** | **N (%)** | **N (%)** |
| **Age group** | | | | | | |
| 2–14 years | 21 (67.7) | 13 (81.3) | 14 (82.4) | 17 (81.0) | 13 (81.3) | 78 (77.2) |
| 15–29 years | 10 (32.3) | 1 (6.3) | 1 (5.9) | 1 (4.8) | 3 (18.8) | 16 (15.8) |
| 30+ years | 0 | 2 (12.5) | 2 (11.8) | 3 (14.3) | 0 | 7 (6.9) |
| **Sex** | | | | | | |
| Male | 19 (61.3) | 13 (81.3) | 10 (58.8) | 11 (52.4) | 9 (56.3) | 62 (61.4) |
| Female | 12 (38.7) | 3 (18.8) | 7 (41.2) | 10 (47.6) | 7 (43.8) | 39 (38.6) |
| **Ulcer location** | | | | | | |
| Head | 1(3.2) | 0 | 0 | 0 | 0 | 1 (1.0) |
| Forearm | 1 (3.2) | 0 | 0 | 1 (4.8) | 0 | 2 (2.0) |
| Leg | 29 (93.5) | 16 (100) | 17 (100) | 20 (96.2) | 16 (100) | 98 (97.0) |
| **Ulcer duration** | | | | | | |
| < 4 weeks | 12 (38.7) | 7 (43.8) | 7 (41.2) | 5 (23.8) | 5 (31.3) | 36 (35.6) |
| 4–12 weeks | 14 (45.2) | 4 (25.0) | 7 (41.2) | 8 (38.1) | 7 (43.8) | 40 (39.6) |
| >13 weeks | 5 (16.1) | 5 (31.3) | 3 (17.6) | 8 (38.1) | 4 (25.0) | 25 (24.8) |
| **Total** | **31 (100)** | **16 (100)** | **17 (100)** | **21 (100)** | **16 (100)** | **101 (100)** |

(yaws), and 74 (73.3%) were positive for *Haemophilus ducreyi* (HD). Eight (7.9%) samples were negative for all four tests. None of the samples was positive for *Mycobacterium ulcerans*.

No statistically significant association was observed between duration of ulcers and the various ulcer categories, at the time of sample collection. In addition, no association was observed between location of ulcers, age or sex of participants, and any of the organisms detected (Table 3). However, regardless of the skin ulcer causative organism detected, there was a significant association (p = 0.001 Chi Square) between duration of the ulcers and age, with children below 15 years presenting the highest proportion of ulcers across the ulcer duration categories (Table 4).

Analysis of the results across the various individual tests showed occurrence of mixed or single organisms in the respective samples. Five (5.4%), 9 (9.7%), and 10 (10.8%) ulcer samples were classified as CL only, *H. ducreyi* only, and yaws only, respectively, based on detection of the corresponding single ulcer causing organisms. On the other hand, two organisms were

**Table 2. Results of single pathogen PCR test and corresponding ulcer classification.**

| Classification* | Study Communities | | | | | |
|---|---|---|---|---|---|---|
| | **Dawa Akura** | **Ashiabre** | **Keri** | **Sibi Hilltop** | **Obunja** | **Total** |
| | **N (%)** | **N (%)** | **N (%)** | **N (%)** | **N (%)** | **N (%)** |
| CL | 9 (29.0) | 4 (25.0) | 16 (94.1) | 4 (19.0) | 7 (43.8) | 40 (39.6) |
| Yaws | 20 (64.5) | 14 (87.5) | 9 (52.9) | 13 (61.9) | 12 (75.0) | 68 (67.3) |
| HD | 19 (61.3) | 10 (62.5) | 15 (88.2) | 16 (76.2) | 14 (87.5) | 74 (73.3) |
| BU | 0 (0) | 0 (0) | 0 (0) | 0 (0) | 0 (0) | 0 (0) |
| Negative | 3 (9.7) | 1 (6.3) | 0 (0) | 3 (14.3) | 1 (6.3) | 8 (7.9) |
| **Samples tested** | **31 (100)** | **16 (100)** | **17 (100)** | **21 (100)** | **16 (100)** | **101 (100)** |

CL-Cutaneous leishmaniasis; HD-Haemophilus ducreyi; BU- Buruli ulcer

**Table 3. Characteristics of pathogens detected.**

| Characteristic | Pathogens identified | | | | | | Samples tested |
|---|---|---|---|---|---|---|---|
| | *Leishmania* | | *H. ducreyi* | | *T. pallidum subsp pertenue* | | |
| | N (%) | p value | N (%) | p value | N (%) | p value | N (%) |
| **Age group (Years)** | | | | | | | |
| 2–14 | 30 (75.0) | 0.279 | 60 (81.1) | 0.693 | 51 (75.0) | 0.251 | 74 (79.6) |
| 15–29 | 6 (15.0) | | 10(13.5) | | 12 (17.7) | | 14 (15.1) |
| 30+ | 4 (10.0) | | 4 (5.4) | | 5 (7.4) | | 5 (5.4) |
| **Sex** | | | | | | | |
| Male | 25 (62.5) | 0.577 | 44 (59.5) | 0.192 | 42 (61.8) | 0.521 | 58 (62.4) |
| Female | 15 (37.5) | | 30 (40.5) | | 26 (38.2) | | 35 (37.6) |
| **Ulcer location** | | | | | | | |
| Head | 0 (0) | 0.671 | 0 | 0.237 | 1 (1.5) | 0.614 | 1 (1.1) |
| Forearm | 1 (2.5) | | 2 (2.7) | | 1 (1.5) | | 2 (2.2) |
| Leg | 39 (97.5) | | 72 (97.3) | | 66 (97.1) | | 90 (96.8) |
| **Ulcer duration** | | | | | | | |
| < 4 weeks | 16 (40.0) | 0.864 | 26 (35.1) | 0.853 | 24 (35.3) | 0.999 | 34 (36.6) |
| 4–12 weeks | 14 (35.0) | | 28 (37.8) | | 24 (35.3) | | 35 (37.6) |
| >13 weeks | 10 (25.0) | | 20 (27.0) | | 20 (29.4) | | 24 (25.8) |
| **Total** | **40 (100)** | | **74 (100)** | | **68 (100)** | | **93 (100)** |

**Table 4. Association between ulcer duration and age of participants.**

| Ulcer Duration (weeks) | age group (Years) | | | | p value* |
|---|---|---|---|---|---|
| | 2–14 N (%) | 15–29 N (%) | 30+ N (%) | Total N (%) | |
| < 4 | 32(43.2) | 2(14.3) | 0(0) | 34(35.0) | 0.001 |
| 4–12 | 27(36.5) | 8 (57.1) | 0(0) | 35 (40.0) | |
| >13 | 15(20.3) | 4(28.6) | 5 (100) | 24 (25.0) | |
| **Total** | **74 (100)** | **14 (100)** | **5 (100)** | **93 (100)** | |

*Chi square test of association

observed in some of the ulcers leading to the following categorization: CL and yaws (4.3%), CL and *H. ducreyi* (11.8%), yaws and H. ducreyi (36.6%). In twenty (19.8%) of the samples, three (*Leishmania, Treponema pallidum* sub species *pertenue, and H. ducreyi)* out of the four organisms investigated, were present simultaneously and were thus classified as CL, yaws, and *H. ducreyi* (Table 5). Fig 3 shows images of the various classifications of the ulcer groups observed. There was no significant difference between the average ulcer sizes and the various ulcer categorizations, based on the number of organisms detected (Table 6).

## Discussion

This study detected single and mixed occurrences of *Leishmania, Treponema pallidum* subsp *pertenue*, and *H. ducreyi* (known causes of leishmaniasis, yaws, and *H. ducreyi*) in ulcers sampled from a recently confirmed foci of cutaneous leishmaniasis (CL) in Ghana. It also confirmed the epidemiological profile of the endemic skin infectious ulcers in rural tropical regions of Africa and the South Pacific [8, 12].

**Table 5. Single or multiple skin ulcer pathogen distribution across study communities.**

| Infections/Colonizations | Study Communities | | | | | Total |
|---|---|---|---|---|---|---|
| | Dawa Akura | Ashiabre | Keri | Sibi Hilltop | Obunja | |
| | N (%) | N (%) | N (%) | N (%) | N (%) | N (%) |
| yaws only | 4 (14.3) | 4 (26.7) | 0 (0) | 2 (11.1) | 0 (0) | 10 (10.8) |
| CL only | 2 (7.1) | 1 (6.7) | 1 (5.9) | 0 (0) | 1 (6.7) | 5 (5.4) |
| HD only | 4 (14.3) | 0 (0) | 0 (0) | 3 (16.7) | 2 (13.3) | 9 (9.7) |
| CL, yaws, and HD | 2 (7.1) | 3 (20.0) | 7 (41.2) | 2 (11.1) | 6 (40.0) | 20 (21.5) |
| yaws and HD | 11 (39.3) | 7 (46.7) | 1 (5.9) | 9 (50.0) | 6 (40.0) | 34 (36.6) |
| CL and yaws | 3 (10.7) | 0 (0) | 1 (5.9) | 0 (0) | 0 (0) | 4 (4.3) |
| CL and HD | 2 (7.1) | 0 (0) | 7 (41.2) | 2 (11.1) | 0 (0) | 11 (11.8) |
| **Total** | **28 (100)** | **15 (100)** | **17 (100)** | **18 (100)** | **15 (100)** | **93 (100)** |

The ulcers were mainly in children and youths under 17 years, with male predominance. Although *Leishmania* infection and subsequent leishmaniasis disease generally tends to be influenced by factors associated with the host, the parasite, as well as the disease vectors, the prevalence of CL in endemic areas usually increases with age till about 15 years. It is assumed

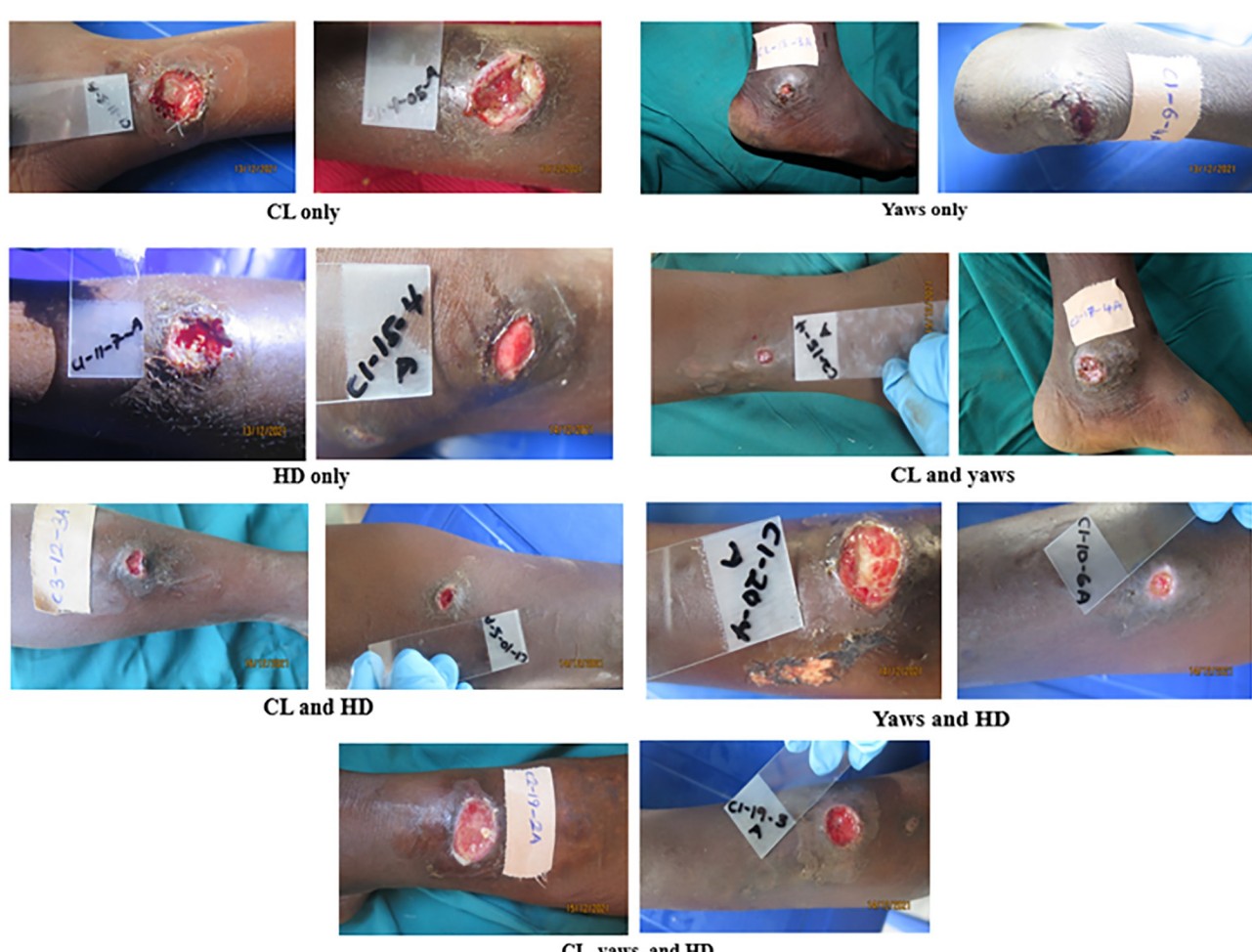

**Fig 3. Pictures of various classifications of skin ulcer groups detected.** Reprinted under a CC BY license, with permission from Dr Richard Akuffo.

**Table 6. Average size of skin ulcer categories.**

| Ulcer category* | Frequency | Average size (cm) n (95% CI) | P value** |
|---|---|---|---|
| Yaws only | 10 | 1.2 (0.9–1.4) | 0.1550 |
| CL only | 5 | 1.6 (1.1–2.1) | |
| HD only | 9 | 1.2 (0.9–1.5) | |
| CL, Yaws, and HD | 20 | 1.9 (1.5–2.3) | |
| Yaws and HD | 34 | 1.8 (1.2–2.3) | |
| CL and Yaws | 4 | 1.4 (0.9–1.9) | |
| CL and HD | 11 | 1.9 (1.1–2.7) | |
| **Total** | **93** | | |

*CL-Cutaneous leishmaniasis; HD-*Haemophilus ducreyi*;

** The Kruskal Wallis H test

that the prevalence of CL levels of at about 15 years because persons exposed early on in life to *Leishmania* infection may have acquired some level of immunity to the infection by then [37, 38].

The ulcers observed in this study were usually single and of small size (about 2 cm) and mostly located on the lower limb. It was not possible to discriminate the skin ulcer aetiologies based on the clinical presentations only, emphasizing the important role for laboratory confirmation and development of laboratory capacities for such diagnosis.

The findings from this study suggest that CL remains highly prevalent in the study areas with coinfections/colonization with *Treponema pallidum* sub species *pertenue* and or *H ducreyi*. Given the reported clinical similarities in the presentation of yaws, CL and *H ducreyi* ulcer, it may therefore be worth considering screening for CL in yaws endemic areas which continue to report large proportions of un-explained yaws-like ulcers, in addition to investigating *H ducreyi* [9, 12].

The typical description of skin ulcers has been based on the detection of the known causative organism. As a result, detection of instances of single occurrences of *Leishmania, Treponema pallidum* sub. Sp. *Pertenue*, and *H. ducreyi* in skin ulcers would have automatically led to their classification as CL, yaws, or *H. Ducreyi* respectively. Other recent studies have confirmed dual occurrences/infection of *Treponema pallidum* sub. Sp. *Pertenue*, and *H. ducreyi* in skin ulcers sampled mainly from yaws endemic communities [6, 9, 13].

Detection of ulcers which could have been classified as CL, yaws, or *H. ducreyi* at the same time, in this study, highlights a potential challenge to skin ulcer diagnosis especially in areas with multiple skin ulcer causing pathogens and yet similar clinical presentation of ulcers [15, 16].

The findings from this study therefore support WHO's call for integrated approach to skin diseases research and control [4, 39]. Success of this strategy also requires determining the unknown cause(s) of cutaneous ulcers (CUs). Etiological characterization of the Cutaneous Ulcer Syndrome in Papua New Guinea using shotgun metagenomics showed multiple bacterial species in the ulcer microbiome. Overall, 471 different species were detected in at least one sample, showing great microbial diversity in ulcer samples. The shotgun metagenomics analysis of the CUs suggests that this disease complex is comprised of multiple distinct bacteriological entities including several mono- or poly-microbial gram-positive infections, in addition to the previously well-recognized *T. pallidum* and *H. ducreyi* infections. The most frequently identified gram positive bacteria were–*S. dysgalactiae, C. diphtheriae, A. haemoliticum,* and *S. aureus*. Clustering analysis revealed at least eight compositional clusters [40].

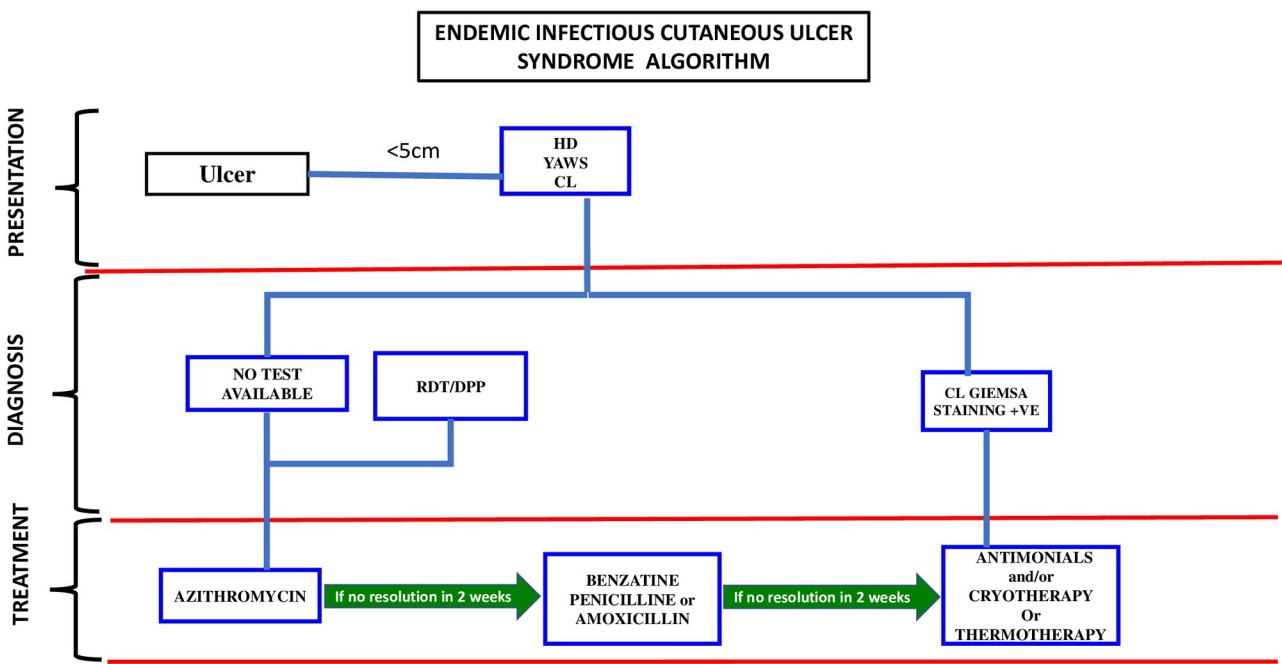

**Fig 4. Proposed algorithm for diagnosis and treatment of endemic infectious cutaneous ulcer disease.** Reprinted under a CC BY license, with permission from Dr. Richard Akuffo. CL—Cutaneous Leishmaniasis; RDT—Rapid Diagnostic Test for yaws; HD—*Haemophilius ducrei* cutaneous ulcer.

A challenge of the microbiologic evaluation described above is the ability to distinguish between bacterial invasion and colonization. Bacteria in ulcers usually act along a continuum from contamination through colonization to critical colonization and finally to infection [41].

Yaws and *H. ducreyi* infections for instance were thought to be exclusively transmitted through non-intact skin contact with infectious lesions. However, *H. ducreyi* DNA has been found in skin swabs from 20% of asymptomatic children and 3/6 bed sheets from the houses of children with ulcers. A high proportion of flies collected in these houses had detectable *H. ducreyi* DNA and half carried both *H. ducreyi* and *T. p. pertenue* DNA. Skin cultures obtained from two asymptomatic children yielded viable *H. ducreyi*, suggesting colonization and a potential reservoir of infection [42].

We suggest the term of Endemic Infectious Cutaneous Ulcers Syndrome (EICUS) to refer to these types of ulcers with multiple skin ulcer causing organisms. Given the interaction of multiple etiologic factors in these ulcers, we propose a syndromic strategy algorithm, which needs to be validated to inform the development of a comprehensive guideline for diagnosing and treating tropical ulcers in the study area (Fig 4). Other studies have also proposed syndromic strategy algorithms to reflect the peculiarities of their study settings [12]. We hope the validation of our proposed algorithm will add to the body of evidence for alleviating the burden of these EICUS.

## Conclusions

The findings of this cross-sectional research provide an overview of the epidemiology of endemic tropical ulcers in the Oti Region of Ghana. This study demonstrated that this ulcerative syndrome is not a "one etiologic infectious disease" but a complex interaction between different microorganisms. The proper role as infection or colonization of each infectious agent is

difficult to characterize and needs further studies to understand the pathogenic dynamics of these complex disease.

## Limitations of the study

This study was based on persons who indicated presence of at least one skin ulcer at the time of the study and may be limited by the willingness of individuals with skin ulcers to participate.

## Supporting information

**S1 File. STROBE checklist: Checklist according to the Strengthening the Reporting of Observational studies in Epidemiology (STROBE) guidelines.**
(DOCX)

## Acknowledgments

This study was a collaborative effort between staff of the Noguchi Memorial Institute for Medical Research, University of Ghana (UG-NMIMR), Instituto de Salud Carlos III, Spain, World Health Organization, Ghana Health Service, University of Ghana Medical School (UGMS), the Kumasi Centre for Collaborative Research (KCCR), the U.S Naval Medical Research Unit #3, Ghana Detachment (NAMRU-3), and La Rabta Hospital Dermatology Department, Research Laboratory, LR18SP01, Faculty of Medecine, University al-Manar 2, Tunis, Tunisia. Additional field support from the University of Ghana Medical school especially from Professor Margaret Lartey, and Dr. Paa Gyasi Hagan is appreciated. Dr. Naiki Attram, Ms. Clara Yeboah, Mr. Seth Addo, and Mr. Mba Mosore of the NAMRU-3 provided significant logistical and technical support toward the research. We are also grateful for the technical support provided by Ms. Shirley Simpson in the area of sample processing for yaws, and *H.ducreyi*. This work would not have been successful without the support and cooperation of members and volunteers from the study communities and local district health facilities. Special mention is made of Mr. Emmanuel Agbodogri, Mr. Bismark Appiah, Mr. Abdulai Mohammed, Ms. Mavis Awasan, Mr. Sylvester Nyaku and Mrs Cynthia Hagan (WCO) for their active involvement in various aspects of the field work. We also appreciate the support of the district and regional health directorates in charge of the study areas.

## Author Contributions

**Conceptualization:** Richard Adjei Akuffo, Yaw Ampem Amoako, Richard Odame Phillips, Michael Wilson, Kingsley Asiedu, Jose-Antonio Ruiz-Postigo, Javier Moreno, Mourad Mokni.

**Formal analysis:** Richard Adjei Akuffo, Yaw Ampem Amoako, Richard Odame Phillips, Javier Moreno, Mourad Mokni.

**Investigation:** Richard Adjei Akuffo, Carmen Sanchez, Ivy Amanor, Jennifer Seyram Amedior, Nana Konama Kotey, Thomas Azurago, Anthony Ablordey, Felicia Owusu-Antwi, Abate Beshah, Javier Moreno, Mourad Mokni.

**Methodology:** Richard Adjei Akuffo, Carmen Sanchez, Ivy Amanor, Jennifer Seyram Amedior, Francis Anto, Anthony Ablordey, Michael Wilson, Kingsley Asiedu, Jose-Antonio Ruiz-Postigo, Javier Moreno.

**Project administration:** Richard Adjei Akuffo, Felicia Owusu-Antwi, Mourad Mokni.

**Resources:** Richard Adjei Akuffo, Nana Konama Kotey, Thomas Azurago, Anthony Ablordey, Felicia Owusu-Antwi, Abate Beshah, Yaw Ampem Amoako, Richard Odame Phillips, Kingsley Asiedu, Javier Moreno.

**Software:** Jose-Antonio Ruiz-Postigo.

**Supervision:** Francis Anto, Anthony Ablordey, Michael Wilson.

**Validation:** Anthony Ablordey.

**Writing – original draft:** Richard Adjei Akuffo.

**Writing – review & editing:** Richard Adjei Akuffo, Carmen Sanchez, Ivy Amanor, Jennifer Seyram Amedior, Nana Konama Kotey, Francis Anto, Thomas Azurago, Anthony Ablordey, Felicia Owusu-Antwi, Abate Beshah, Yaw Ampem Amoako, Richard Odame Phillips, Michael Wilson, Kingsley Asiedu, Jose-Antonio Ruiz-Postigo, Javier Moreno, Mourad Mokni.

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
