## [Decision Letter · Decision Letter 0]

18 Jul 2023

PONE-D-23-14994Endemic infectious cutaneous ulcers syndrome in the Oti Region of Ghana: study of cutaneous leishmaniasis, yaws and Haemophilus ducreyiPLOS ONE

Dear Dr. Akuffo,

Thank you for submitting your manuscript to PLOS ONE. After careful consideration, we feel that it has merit but does not fully meet PLOS ONE’s publication criteria as it currently stands. Therefore, we invite you to submit a revised version of the manuscript that addresses the points raised during the review process.

We look forward to receiving your revised manuscript.

Kind regards,

Alireza Badirzadeh

Academic Editor

PLOS ONE

3. Please include a caption for figure 1.

4. We note that Figures 1 and 3 in your submission contain copyrighted images. All PLOS content is published under the Creative Commons Attribution License (CC BY 4.0), which means that the manuscript, images, and Supporting Information files will be freely available online, and any third party is permitted to access, download, copy, distribute, and use these materials in any way, even commercially, with proper attribution. For more information, see our copyright guidelines: http://journals.plos.org/plosone/s/licenses-and-copyright.

1. You may seek permission from the original copyright holder of Figures 1 and 3 to publish the content specifically under the CC BY 4.0 license.

Reviewers' comments:

Reviewer's Responses to Questions

**Comments to the Author**

1. Is the manuscript technically sound, and do the data support the conclusions?

Reviewer #1: Yes

Reviewer #2: Yes

2. Has the statistical analysis been performed appropriately and rigorously? 

Reviewer #1: Yes

Reviewer #2: Yes

3. Have the authors made all data underlying the findings in their manuscript fully available?

Reviewer #1: Yes

Reviewer #2: Yes

4. Is the manuscript presented in an intelligible fashion and written in standard English?

Reviewer #1: Yes

Reviewer #2: Yes

5. Review Comments to the Author

Reviewer #1: Dear Editor,

Thank you for letting me to review the manuscript “Endemic infectious cutaneous ulcers syndrome in the Oti Region of Ghana: study of cutaneous leishmaniasis, yaws and Haemophilus ducreyi”. In this study, Akuffo et al investigated skin ulcers with unknown causes in the Oti region of Ghana. They identified single and mixed occurrence of causative organisms of CL, Yaws and H. ducreyi in wounds of patients from Oti region. Considering that each disease requires its own prescription, they emphasized on expansion of a comprehensive guideline for the diagnosis and treatment of tropical ulcers in the study areas. Their study can be valuable in the sense that the clinical features of similar wounds require different diagnosis and treatment methods. Overall, this study is relatively well designed, conducted and written.

You can find my comments on improving this manuscript below.

1- Titles

Long title: The causes of diseases are not written uniformly, often with generic names and sometimes with scientific names.

Suggested short title: Infectious skin ulcer syndrome in Oti region

2- The lack of uniformity that was suggested about the title also applies to keywords

3- Please remove the word “Leishmania” from the Key words.

4- The text of the legend of the tables and images need to be improved

5- It is not customary to include pictures and tables in the discussion. Eg. L:339

6- The use of the word microbiome in line 345 is incorrect

7- A very long Acknowledgements part

Reviewer #2: Dear authors,

First and foremost, I would like to congratulate you on addressing an important issue in the African continent. The manuscript is commendable and can be published in its current state. However, I believe the following comments can further enhance the quality of the article.

1. Please explain employed molecular methods, briefly.

2. Please provide your molecular results using images.

3. In certain sections of the discussion, like line 289, you merely stated the outcome without providing an accompanying explanation regarding its potential underlying cause. Kindly revise those particular segments by incorporating an explanation for the observed result.

Sincerely

6. PLOS authors have the option to publish the peer review history of their article (what does this mean?). If published, this will include your full peer review and any attached files.

Reviewer #1: No

Reviewer #2: No

---

## [Author Response · Author response to Decision Letter 0]

9 Aug 2023

Academic editor

Comment 1: Please include a caption for figure 1.

We note that Figures 1 and 3 in your submission contain copyrighted images. All PLOS content is published under the Creative Commons Attribution License (CC BY 4.0), which means that the manuscript, images, and Supporting Information files will be freely available online, and any third party is permitted to access, download, copy, distribute, and use these materials in any way, even commercially, with proper attribution. For more information, see our copyright guidelines: http://journals.plos.org/plosone/s/licenses-and-copyright.

1. You may seek permission from the original copyright holder of Figures 1 and 3 to publish the content specifically under the CC BY 4.0 license.

In the figure caption of the copyrighted figure, please include the following text: “Reprinted from [ref] under a CC BY license, with permission from [name of publisher], original copyright [original copyright year].

Response: The Figure 1 previously included in the manuscript was obtained from a collaborator. However, that Figure 1 has been removed from the manuscript due to non-response from the collaborator for permission to publish the figure.

A new Figure 1 has been included based on results from this study. As corresponding author, I have granted permission for its publication under the Creative Commons Attribution License (CCAL) CC BY 4.0.

Figure 3 was generated from the study being reported and is not based on copyright work of another author. As corresponding author for this submission, I have also granted permission for publication of Figure 3. All other Figures indicated in this manuscript are based on or derived from data obtained from the study being reported. I have therefore granted permission for their publication under the Creative Commons Attribution License (CCAL) CC BY 4.0. I have therefore indicated same in the figure captions.

Comment 2: Please review your reference list to ensure that it is complete and correct. If you have cited papers that have been retracted, please include the rationale for doing so in the manuscript text, or remove these references and replace them with relevant current references. Any changes to the reference list should be mentioned in the rebuttal letter that accompanies your revised manuscript. If you need to cite a retracted article, indicate the article’s retracted status in the References list and also include a citation and full reference for the retraction notice.

Response: The reference list has been checked. I can confirm that it is complete and correct.

Feedback from Reviewers:

Reviewer 2

Comment:1- Titles. Long title: The causes of diseases are not written uniformly, often with generic names and sometimes with scientific names.

Suggested short title: Infectious skin ulcer syndrome in Oti region

Response: The authors have accepted the proposal for short title and have accordingly amended same at Line 5. 

Regarding the causes of diseases not being uniformly written, the authors indicate that their preference was to use the generic names of the skin diseases in the title. In the case of Haemophilus ducreyi cutaneous ulcers however, there is no generic name yet for it beside Haemophilus ducreyi cutaneous ulcer. The authors have therefore changed the text from just ‘Haemophilus ducreyi’ to ‘Haemophilus ducreyi cutaneous ulcer’ to better represent it and to avoid the confusion between Haemophilus ducreyi and Haemophilus ducreyi ulcer (line 3).

Comment 2- The lack of uniformity that was suggested about the title also applies to keywords

Comment 3- Please remove the word “Leishmania” from the Key words.

Response: The authors accept the suggestion. The word Leishmania has been removed from the key words (line 67).

Comment 4- The text of the legend of the tables and images need to be improved

Response: The text of the legend of the tables and images have been generally improved.

Comment 5- It is not customary to include pictures and tables in the discussion. Eg. L:339

Response: The authors agree with the assertion. However, the authors are introducing an algorithm based on a discussion of the results from the study. This explains the introduction of figure 4 at the discussion section.

Comment 6- The use of the word microbiome in line 345 is incorrect

Response: The authors agree with this comment. The text at line 345 has accordingly been amended.

Comment 7- A very long Acknowledgements part

Response: The authors accept the comment. Effort has been made to reduce the length of the acknowledgements section (Lines 421-425, 432, 436-440)

Reviewer #2: 

Comment 1: Please explain employed molecular methods, briefly.

Response: Molecular methods employed have been better explained (Lines 200-210; 212-221; 225-233)

Comment 2. Please provide your molecular results using images.

Response: A new figure 1 has been added to present visualization of nested PCR (Ln-PCR) amplification products on 1.5% agarose gel based on tests for Leishmania infection.

Comment 3. In certain sections of the discussion, like line 289, you merely stated the outcome without providing an accompanying explanation regarding its potential underlying cause. 

Response: The authors accept the feedback from reviewer. Lines 319 to 324 capture additional explanation provided by authors.

---

## [Decision Letter · Decision Letter 1]

11 Sep 2023

Endemic infectious cutaneous ulcers syndrome in the Oti Region of Ghana: study of cutaneous leishmaniasis, yaws and Haemophilus ducreyi cutaneous ulcers

PONE-D-23-14994R1

Dear Dr. Richard Adjei Akuffo,

We’re pleased to inform you that your manuscript has been judged scientifically suitable for publication and will be formally accepted for publication once it meets all outstanding technical requirements.

Kind regards,

Alireza Badirzadeh

Academic Editor

PLOS ONE

Additional Editor Comments (optional):

Reviewers' comments:

Reviewer's Responses to Questions

**Comments to the Author**

1. If the authors have adequately addressed your comments raised in a previous round of review and you feel that this manuscript is now acceptable for publication, you may indicate that here to bypass the “Comments to the Author” section, enter your conflict of interest statement in the “Confidential to Editor” section, and submit your "Accept" recommendation.

Reviewer #1: All comments have been addressed

Reviewer #2: All comments have been addressed

2. Is the manuscript technically sound, and do the data support the conclusions?

Reviewer #1: Partly

Reviewer #2: Yes

3. Has the statistical analysis been performed appropriately and rigorously? 

Reviewer #1: N/A

Reviewer #2: Yes

4. Have the authors made all data underlying the findings in their manuscript fully available?

Reviewer #1: Yes

Reviewer #2: (No Response)

5. Is the manuscript presented in an intelligible fashion and written in standard English?

Reviewer #1: Yes

Reviewer #2: Yes

6. Review Comments to the Author

Reviewer #1: The authors have largely applied my comments in their manuscript. While they have done this study with limited facilities and in difficult conditions

Reviewer #2: Dear authors;

First, I appreciate you providing information about the etiology of cutaneous ulcers in Ghana, and your manuscript is desired to be accepted.

All the best

7. PLOS authors have the option to publish the peer review history of their article (what does this mean?). If published, this will include your full peer review and any attached files.

Reviewer #1: No

Reviewer #2: No

---

## [Editor Report · Acceptance letter]

18 Sep 2023

PONE-D-23-14994R1 

Endemic infectious cutaneous ulcers syndrome in the Oti Region of Ghana: study of cutaneous leishmaniasis, yaws and *Haemophilus ducreyi* cutaneous ulcers 

Dear Dr. Akuffo:

I'm pleased to inform you that your manuscript has been deemed suitable for publication in PLOS ONE. Congratulations! Your manuscript is now with our production department. 

Kind regards, 

on behalf of

Dr. Alireza Badirzadeh 

Academic Editor

PLOS ONE